# Hybrid Policy Optimization from Imperfect Demonstrations

**Hanlin Yang**
Sun Yat-sen University

**Chao Yu**[*]
Sun Yat-sen University

**Peng Sun**
ByteDance

**Siji Chen**
Sun Yat-sen University

yuchao3@mail.sysu.edu.cn

## Abstract

Exploration is one of the main challenges in Reinforcement Learning (RL), especially in environments with sparse rewards. Learning from Demonstrations (LfD) is a promising approach to solving this problem by leveraging expert demonstrations. However, expert demonstrations of high quality are usually costly or even impossible to collect in real-world applications. In this work, we propose a novel RL algorithm called HYbrid Policy Optimization (HYPO), which uses a small number of imperfect demonstrations to accelerate an agent's online learning process. The key idea is to train an offline guider policy using imitation learning in order to instruct an online agent policy to explore efficiently. Through mutual update of the guider policy and the agent policy, the agent can leverage suboptimal demonstrations for efficient exploration while avoiding the conservative policy caused by imperfect demonstrations. Empirical results show that HYPO significantly outperforms several baselines in various challenging tasks, such as MuJoCo with sparse rewards, Google Research Football, and the AirSim drone simulation.

## 1 Introduction

Reinforcement Learning (RL) (Sutton & Barto, 2018) plays an important role in solving real-world control problems with large state-action spaces. In RL, an agent learns a decision-making policy through interaction with the environment, based on a reward function that indicates the agent's learning goal. However, pre-defining a reward function relies heavily on human knowledge that varies from one to another, and sometimes carefully hand-tuned reward function could lead to undesired or even hazardous policies (Devidze et al., 2021). Therefore, a more intuitive way of reward definition is to utilize the completeness of the task, e.g., the distance that a robot walks or scoring a goal in football games. While being unbiased towards the learning goal, learning with such sparse rewards could be challenging due to the inefficient exploration in the large problem space.

In recent years, there has been intensive research interest in Learning from Demonstrations (LfD) as a promising approach to addressing sparse reward tasks by leveraging the expert demonstrations. Methods like DDPGfD (Vecerík et al., 2017), DQfD (Hester et al., 2018) and AWAC (Nair et al., 2020) normally require that the demonstrations consist of complete trajectories including state, actions and rewards, and thus are not applicable in real-world applications when it is only possible to observe the state and action information in the demonstrations. Other methods based on Imitation Learning (IL), such as Generative Adversarial Imitation Learning (GAIL) (Ho & Ermon, 2016) and Adversarial Inverse Reinforcement Learning (AIRL) (Fu et al., 2017), usually require that the demonstrations be generated by an optimal expert, which are difficult to achieve in complex tasks. More recent approaches such as Policy Optimization with Demonstrations (POfD) (Kang et al.,

---

[*]Corresponding author. Code is available at https://github.com/joenghl/HYPO.

37th Conference on Neural Information Processing Systems (NeurIPS 2023).

2018) and Learning Online with Guidance Offline (LOGO) (Rengarajan et al., 2022) can result in an excessively conservative policy (Nair et al., 2020), due to the negative impact of imperfect demonstrations. Specifically, when the agent performs better than the suboptimal expert policy, simply using the demonstration-guided reward or distilling knowledge (e.g., POfD and LOGO) from the suboptimal policy will cause an excessively conservative policy of the online learning agent.

In this paper, we propose HYbrid Policy Optimization (HYPO), a novel RL algorithm that is capable of achieving near-optimal performance in sparse reward environments using only a small number of imperfect demonstrations with low quality and incomplete trajectories. The basic idea is to learn an offline imitation policy (i.e., the guider) to guide the learning of the agent and an online learning policy (i.e., the agent) to interact with the environment. These two policies are updated mutually to help the agent learn from both the expert demonstrations and the environment more efficiently. The key insight of HYPO is that, the guider imitates the expert to guide the learning of the agent during the initial stage, and then the agent learns to outperform the expert through online interactions with the environment. We evaluate HYPO on a wide variety of control tasks, including MuJoCo with sparse rewards, Google Research Football and AirSim UAV simulation. Besides, we also investigate how the number and quality of the demonstrations can influence the performance of HYPO and other baseline methods. The results show that 1) HYPO can learn successful policies in these challenging tasks efficiently, and 2) HYPO is minimally affected by the number and quality of the demonstrations.

## 2 Related Work

### 2.1 Offline RL

Offline RL (Levine et al., 2020) enables to learn policies using only offline data, without any online interaction with the environment. However, the related offline RL methods, such as BCQ (Fujimoto et al., 2019), BEAR (Kumar et al., 2019), BRAC (Wu et al., 2019), and Fisher-BRC (Kostrikov et al., 2021) still suffer from the issue of distributional shift, i.e., the policy is trained under one distribution but evaluated on a different distribution. A common way to mitigate this issue is to restrict the policy distribution to be similar to the policy that generates the offline data. However, this constraint can result in excessively conservative policies. In contrast, HYPO can avoid the distribution shift problem since the offline trained guider has no interaction with the environment, but is only used to guide the online agent for more efficient exploration. In addition, unlike the existing offline RL approaches that normally require a large number of data with complete trajectories, thus have limited applications when collecting a large number of perfect demonstrations is either time-consuming or even infeasible, HYPO is able to learn a near-optimal policy given only a small number of incomplete demonstrations generated by a suboptimal policy.

### 2.2 IL

IL methods are designed to mimic the behaviors of experts. As one of the most well-known IL algorithms, Behavior Cloning (BC) learns a policy by directly minimizing the discrepancy between the agent and the expert in the demonstration data. However, BC has been shown to suffer from issues such as compounding errors and the inability to handle the distributional shift Ross et al. (2011). As a recent BC-based method, DWBC Xu et al. (2022) trains a pure offline BC task to attain the expert performance by using a discriminator that distinguishes the optimal expert data and the suboptimal data. Different from DWBC, HYPO can learn policies that outperform the expert by using online interaction with the environments and offline guidance from the guider. Moreover, unlike BC-based methods that always require optimal demonstrations, HYPO is capable of learning with only imperfect demonstrations. As another type of IL, Inverse RL (IRL) (Ng et al., 2000) learns a reward function that explains the expert behavior and then uses this reward function to guide the agent's learning process. Some popular approaches use adversarial training to train a policy that is similar to the expert policy, but at the same time is robust to distributional shift, such as the GAIL (Ho & Ermon, 2016) and AIRL (Fu et al., 2017). However, unlike HYPO, all these methods assume that the expert actions be optimal.

## 2.3 LfD

The goal of LfD is to use demonstration data to aid agent online learning, so as to overcome the exploration problem in RL, especially in environments with sparse rewards Schaal (1996). DQfD (Hester et al., 2018) and DDPGfD (Vecerík et al., 2017) rely on complete offline expert demonstrations with associated rewards to accelerate the online learning process in discrete and continuous action spaces, respectively. POfD Kang et al. (2018) combines the benefits of both imitation learning and RL to learn policies from both demonstration data and online interactions by learning from a shaped reward averaged over the original environment reward and a demonstration-guided reward. However, as the environment reward is much sparser than the demonstration-guided reward, such a mixed reward may cause the learning objective to deviate from the goal of the optimal policy. Another LfD approach is LOGO (Rengarajan et al., 2022), which pre-trains an expert policy to guide the learning of agent, and uses a trust region controlled by a hyper-parameter to restrict agent learning. Unlike LOGO that the final policy may be excessively conservative due to the fixed suboptimal expert, HYPO enables the agent to outperform the expert through updating the guider and the agent mutually.

## 3 Preliminaries

### 3.1 Markov Decision Process

We consider the standard Markov Decision Process (MDP) (Sutton & Barto, 2018) as the mathematical framework for modeling sequential decision-making problems, which is defined by a tuple $\langle \mathcal{S}, \mathcal{A}, P, r, d_0, \gamma \rangle$, where $\mathcal{S}$ is a finite set of states, $\mathcal{A}$ is a finite set of actions, $P : \mathcal{S} \times \mathcal{A} \times \mathcal{S} \to \mathbb{R}$ is the transition probability function, $r : \mathcal{S} \to \mathbb{R}$ is the reward function, $d_0$ is the initial distribution, and $\gamma \in [0, 1)$ is the discount factor. An agent interacts with the environment based on policy $\pi : \mathcal{S} \times \mathcal{A} \to [0, 1]$, which maps from state to distribution over actions. The performance of $\pi$ is evaluated by its expected discounted reward $J(\pi)$:

$$J(\pi) = \mathbb{E}_\pi \left[ r(s, a) \right] = \mathbb{E}_{(s_0, a_0, s_1, \ldots,)} \left[ \sum_{t=0}^{\infty} \gamma^t r\left( s_t, a_t \right) \right], \tag{1}$$

where $(s_0, a_0, s_1, \ldots)$ is a trajectory generated from interaction with the environment, i.e., $s_0 \sim d_0$, $a_t \sim \pi(\cdot | s_t)$ and $s_{t+1} \sim P(\cdot | s_t, a_t)$. The value function of a policy is defined as $V_\pi(s) = \mathbb{E}_{\tau \sim \pi} \left[ \sum_{t=0}^{\infty} \gamma^t r(s_t, a_t) | s_0 = s \right]$, i.e., the expected cumulative discounted reward obtained by the policy $\pi$ from the state $s$. Correspondingly, the action-value function is defined as $Q_\pi(s, a) = \mathbb{E}_{\tau \sim \pi} \left[ \sum_{t=0}^{\infty} \gamma^t r(s_t, a_t) | s_0 = s, a_0 = a \right]$, and the advantage function is $A_\pi(s, a) = Q_\pi(s, a) - V_\pi(s)$. The objective of RL algorithm is to discover the optimal policy $\pi^*$ that can maximize the expectation of discounted return $J(\pi)$, i.e., $\pi^* = \mathrm{argmax}_\pi J(\pi)$.

### 3.2 Learning with Offline Demonstrations

BC and GAIL are well-known baseline methods for learning with offline demonstrations. The difference is that BC is a pure offline method without any interactions with the environment, while GAIL is an online learning method through generating and learning from the trajectories.

**BC.** BC is used for matching the distribution over actions, with its objective given by:

$$\mathcal{L}_\pi = \mathbb{E}_{(s,a) \sim \mathcal{D}} \left[ -\log \pi(a|s) \right], \tag{2}$$

where $\mathcal{D}$ is the offline demonstrations. Optimizing objective (2) will make the distribution of actions w.r.t. $\pi_b$ match the one in the expert policy that generates the state-action pairs in $\mathcal{D}$.

**GAIL.** The discriminator aims to distinguish between the offline expert demonstrations and the trajectories sampled by an online agent. The objective of the discriminator in GAIL is given by:

$$\mathcal{L}_{d-\mathrm{GAIL}} = \min_d \mathbb{E}_{(s,a) \sim \mathcal{D}} \left[ -\log d(s, a) \right] + \mathbb{E}_{(s,a) \sim \mathcal{B}} \left[ -\log \left( 1 - d(s, a) \right) \right], \tag{3}$$

where $\mathcal{D}$ is the offline demonstrations, $\mathcal{B}$ in the replay buffer consists of trajectories sampled by the agent, $d$ is the discriminative classifiers $d : \mathcal{S} \times \mathcal{A} \to (0, 1)$. Optimizing objective (3) will

make the learned discriminator assign 1 to all transitions from the offline demonstration $\mathcal{D}$ and 0 to all transitions from $\mathcal{B}$. During training, the discriminator will assign a large reward to agent when $d_{(s,a)\sim\mathcal{B}}(s,a)$ is close to 1, which is inspired by the generative adversarial training.

The discriminator can be interpreted as a new reward function $r(s,a) = -\log\big(1-d(s,a)\big)$ providing learning signal to the agent. In this way, the agent will receive larger reward if its trajectories are more like the expert, leading to against the discriminator.

# 4 Hybrid Policy Optimization with Imperfect Demonstrations

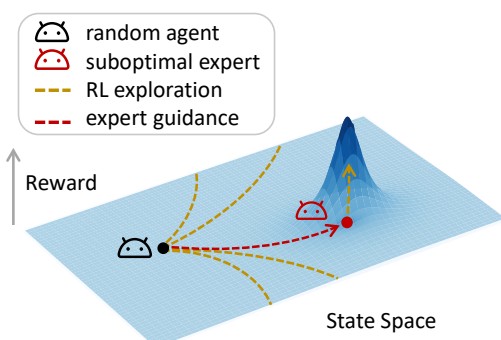

Figure 1: Our proposed method provides a suboptimal expert direction (red curve) for the agent, which prevents it from pointless exploration (brown curves) under sparse reward environments.

Before presenting detailed descriptions of our method, we use Figure 1 to illustrate the overall motivation. In the early learning stage, the agent usually suffers from pointless exploration (i.e., brown curves) due to the sparsity of the reward signal. In such stage, the agent is generally worse than the expert, and thus can gain beneficial guidance from the expert demonstrations to attain a relatively fair performance that is close to the expert. In the later learning stages, we expect the agent to surpass the expert through exploration in the environment, hence being too close to the expert can hinder the policy improvement. However, without constraining the deviation to the expert, the agent can still encounter performance collapse due to the pointless exploration caused by the sparse reward. We tackle these challenges by updating the two policies of the guider and agent mutually, to provide the agent with an appropriate guidance. In this way, the agent is able to outperform the expert to achieve a near-optimal performance.

Our goal is to develop an algorithm that can leverage the imperfect demonstrations generated by a suboptimal expert to boost online learning in sparse reward environment. We assume that there is a suboptimal expert policy $\pi_e$ but we have no access to it. All we can use are its demonstrations, which have the form $\mathcal{D} = \{\tau^i\}_{i=1}^n$, where $\tau^i = (s_1^i, a_1^i, \ldots, s_t^i, a_t^i), \tau^i \sim \pi_e$. And we make the following reasonable and indispensable assumption concerning the caliber of the expert policy:

**Assumption 1.** *In the initial learning stages, the agent can get higher advantage values than current policy $\pi$ if it acts according to the expert policy $\pi_e$,*

$$\mathbb{E}_{a_e \sim \pi_e, a \sim \pi}\left[A_\pi(s, a_e) - A_\pi(s, a)\right] \geq \xi > 0, \forall s \in \mathcal{S} \tag{4}$$

The above assumption implies that taking actions according to the expert will provide a higher advantage than taking actions according to the current policy $\pi$. This is reasonable since the expert policy perform much better than an untrained policy in the initial learning stages.

To better illustrate the analysis and simplify the formulation below, we use the term "expert" to indicate the suboptimal expert unless using the term "optimal" as the prefix. The offline and the online policy is respectively denoted as $\pi_b$ and $\hat{\pi}$. The demonstrations consisting of trajectories generated by the suboptimal expert is denoted as $\mathcal{D}$, while the replay buffer consisting of trajectories sampled by the agent is denoted as $\mathcal{B}$. The output of the descriminator is $d : \mathcal{S} \times \mathcal{A} \rightarrow (0, 1)$.

The three key components in HYPO are the discriminator, the offline guider, and the online agent. The discriminator is used to control the learning objective of the offline guider by distinguishing whether the trajectories are generated by the expert or the agent. The offline guider performs a BC task to provide the agent an adaptive guidance by learning from the expert and the agent dynamically. The online agent policy interacts with the environments and distills knowledge from the guider policy constantly to outperform the expert.

## 4.1 Semi-supervised Discriminator Learning

The discriminator in GAIL mentioned in Subsection 3.2 suffers from the overfitting problem, since the well-learned discriminator may overfit to the suboptimal expert (Zolna et al., 2021). Concretely, as the agent policy $\hat{\pi}$ improves, the behaviors of the agent become increasingly similar to those of the expert. However, due to the suboptimality of the expert, it is crucial to avoid overfitting to the suboptimal expert, as it can hinder the agent from converging to the optimal performance. To address this issue, we formulate the discriminator objective as a positive-unlabeled (PU) reward learning problem (Xu & Denil, 2021), which allows us to train the discriminator by treating the agent trajectories as an unlabeled mixture of the expert trajectories and the agent's trajectories. In this way, the guider can also learn from the agent since the agent's trajectories are not just the negative data, but the mixture of positive and negative data. The formulated discriminator objective is given as follows:

$$\min_d \eta \underset{(s,a)\sim\mathcal{D}}{\mathbb{E}}\Big[-\log d(s,a)\Big] + \underset{(s,a)\sim\mathcal{B}}{\mathbb{E}}\Big[-\log\big(1-d(s,a)\big)\Big] - \eta \underset{(s,a)\sim\mathcal{D}}{\mathbb{E}}\Big[-\log\big(1-d(s,a)\big)\Big], \quad (5)$$

where $\mathcal{D}$ denotes the offline demonstrations, $\mathcal{B}$ the replay buffer consisting of the trajectories sampled by the agent, and $\eta \in [0,1]$ the positive class prior, which is assumed to be known.

Furthermore, it is difficult to distinguish between the expert's and agent's trajectories by only relying on the information of state-action pairs. To mitigate this issue, an additional signal $\log\pi_b$ is added into the input of the discriminator, leading to the final objective of the discriminator as follows:

$$\begin{aligned}
\mathcal{L}_d = \eta \underset{(s,a)\sim\mathcal{D}}{\mathbb{E}}\Big[-\log d(s,a,\log\pi_b)\Big] + \underset{(s,a)\sim\mathcal{B}}{\mathbb{E}}\Big[-\log\big(1-d(s,a,\log\pi_b)\big)\Big] \\
- \eta \underset{(s,a)\sim\mathcal{D}}{\mathbb{E}}\Big[-\log\big(1-d(s,a,\log\pi_b)\big)\Big].
\end{aligned} \quad (6)$$

Specifically, when considering state-action pairs $(s,a)$ originating from the agent's trajectories, $\pi_b(a|s)$ will assign large probabilities to the agent's actions within these corresponding states. Conversely, for state-action pairs $(s,a)$ derived from the expert's trajectories, $\pi_b(a|s)$ will assign small probabilities to the expert's actions within these corresponding states.

**Remark on $\eta$.** It is worth noting that, unlike the standard positive-unlabeled learning setup, the positive class prior $\eta$ in our setting changes as the online policy learning progresses and the distribution of states in the replay buffer evolves. Specifically, $\eta$ is set to increase as the learning progresses.

## 4.2 Adaptive Target for Offline Imitation

The objective of BC task is to mimic the expert policy through matching the conditional distribution $\pi(\cdot|s)$ over actions. Therefore, the BC methods cannot learn a policy that outperforms the expert without additional guidance. In HYPO, $\pi_b$ is able to use the information from the trajectories sampled by the online policy $\hat{\pi}$ to learn from the agent, which can assist $\pi_b$ to attain a better performance than the original expert. Specifically, we use two adaptive functional weights $\mathcal{F}_{\text{Expert}}(d)$ and $\mathcal{G}_{\text{Agent}}(d)$ to determine the learning objective of $\pi_b$ dynamically:

$$\mathcal{L}_{\pi_b} = \underset{(s,a)\sim\mathcal{D}}{\mathbb{E}}\Big[-\log\pi_b(a|s)\cdot\mathcal{F}\big(d(s,a,\log\pi_b)\big)\Big] + \underset{(s,a)\sim\mathcal{B}}{\mathbb{E}}\Big[-\log\pi_b(a|s)\cdot\mathcal{G}\big(d(s,a,\log\pi_b)\big)\Big], \quad (7)$$

where $\mathcal{F}$ and $\mathcal{G}$, denoting $\mathcal{F}_{\text{Expert}}(d)$ and $\mathcal{G}_{\text{Agent}}(d)$, respectively, are the functional weights to determine the objective of $\pi_b$. The functional weights are expected to force the $\pi_b$ to approach the expert in the initial stage, and to increase $\mathcal{L}_d$ to ensure discriminator robust throughout the whole learning process. Since the performance of the online learning policy differs significantly, ranging from being random to reaching the near-optimal, the discriminator needs to be robust enough to handle these changes. Inspired by the adversarial robustness (Carlini et al., 2019) and the weighted discriminator (Xu et al., 2022), we use $\pi_b$ to maximize $\mathcal{L}_d$ to improve the robustness of discriminator by minimizing the worst-case error. Consequently, the form of the functional weights of $\mathcal{F}_{\text{Expert}}$ and $\mathcal{G}_{\text{Agent}}$ can be derived as follows:

$$\mathcal{F}_{\text{Expert}}(d) = \alpha - \frac{\eta}{d(1-d)}, \qquad \mathcal{G}_{\text{Agent}}(d) = \frac{1}{1-d}, \quad (8)$$

where $\alpha$ is a weight factor. Refer to Appendix B for the detailed derivation.

To understand the trend of the functional weight $\mathcal{F}_{\text{Expert}}(d)$ intuitively, we visualize it as illustrated in Figure 2. For transitions from $\mathcal{D}$, the weight is insensitive to $d$ when $\eta$ is small, since the proportion of positive samples in the $(s, a)$ pairs generated by the agent is low. In this case, $\pi_b$ can provide the online agent with guidance by learning towards the expert. As training progresses, the proportion $\eta$ of positive samples in agent trajectories gradually increases, thus $d$ begins to dominate the weight changes of $\mathcal{F}_{\text{Expert}}(d)$. In this case, $\mathcal{F}_{\text{Expert}}(d)$ becomes sensitive to $d$, and gets smaller since the $d$ gets

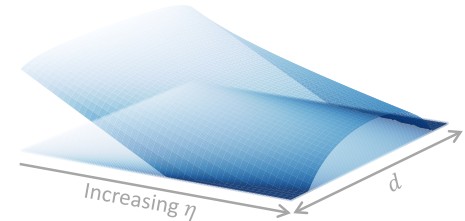

Figure 2: The trend of weight $\mathcal{F}_{\text{Expert}}(d)$. The deeper blue color means the lower weight value. $\eta$ is increasing during the training and $d$ is various from 0.1 to 0.9 (clipped).

large when $(s, a)$ comes from the expert, resulting in the learning towards the agent for $\pi_b$. As for the other weight of $\mathcal{G}_{\text{Agent}}$, which will be large if the $(s, a)$ sampled by the agent $\hat{\pi}$ is similar to that of the expert, changes insensitively when $d$ remains low.

### 4.3 Performance Improvement of Online Learning

An useful reward signal is necessary for online RL algorithms to learn efficiently. However, in the sparse reward environments, the agent cannot explore states that have non-zero reward without additional guidance. To this end, we use an offline guider to provide the agent with guidance, which can significantly boost the exploration of agent, especially in the initial stages of training. Specifically, the online agent policy $\hat{\pi}$, which learns from the sparse environment rewards, constantly distills knowledge from the guider policy $\pi_b$. This is implemented by adding another corrective objective on the original PPO (Schulman et al., 2017) policy update:

$$J_{\hat{\pi}}^{\text{HYPO}}(\theta) = \mathbb{E}_t\Big[\min\Big(r_t(\theta)A_t, \text{clip}\big(r_t(\theta), 1-\epsilon, 1+\epsilon\big)A_t\Big) - CD_{\text{KL}}(\hat{\pi}||\pi_b)\Big], \qquad (9)$$

where $C$ is a decreasing coefficient of the KL-divergence. In Eq (9), we append a constraint of the KL-divergence between the guider policy $\pi_b$ and the agent policy $\hat{\pi}$. In this way the agent can distill knowledge constantly from the guider. Besides, the decreasing coefficient of $C$ can prevent the agent from the excessively conservative policy. Next, we theoretically analyze how the additional constraint can influence the agent. We consider the following two cases: case 1) the agent's performance is worse than the expert, which corresponds to satisfaction of Assumption 1, and case 2) the agent's performance is better than the expert, which means Assumption 1 is not satisfied.

**Case 1.** In the initial stage of training, the offline guider policy $\pi_b$ can be approximately treated as the expert policy since the BC method is more efficient than policy gradient (PG) and the $\mathcal{F}_{\text{Expert}}$ keeps large, which drives $\pi_b$ to update towards the expert. Now the performance improvement guarantee of the online agent under the Assumption 1 can be given as follows:

**Proposition 1.** *Let $\tilde{\pi}$ be a policy that satisfies Assumption 1. Then, for policy $\hat{\pi}$,*

$$J_R(\hat{\pi}) - J_R(\tilde{\pi}) \geq (1-\gamma)^{-1}\xi - (1-\gamma)^{-1}\epsilon_{R,\tilde{\pi}}\sqrt{2D_{\text{KL}}^{\hat{\pi}}(\hat{\pi}, \pi_b)}, \qquad (10)$$

*where $\epsilon_{R,\tilde{\pi}} = \max_{s,a}|A_R^{\tilde{\pi}}(s, a)|$.*

*Proof.* Refer to the Appendix B.

It is reasonable to assume that $\hat{\pi}$ satisfies Assumption 1 in the initial stage of learning since the current policy $\hat{\pi}$ of agent is learning from scratch. Then, minimizing $D_{\text{KL}}^{\hat{\pi}}(\hat{\pi}, \pi_b)$ can get a non-negative lower bound in Equation (10), which means that we can accelerate the online learning of the agent by minimizing the KL-divergence between $\hat{\pi}$ and $\pi_b$. This process can be described as, the agent learns from the guider to attain the expert performance, which can constantly get non-zero rewards from the sparse environments.

**Case 2.** As the training progresses, the agent can achieve the expert performance efficiently. This can be realized in most of the previous imitation methods. However, we want to investigate that, how

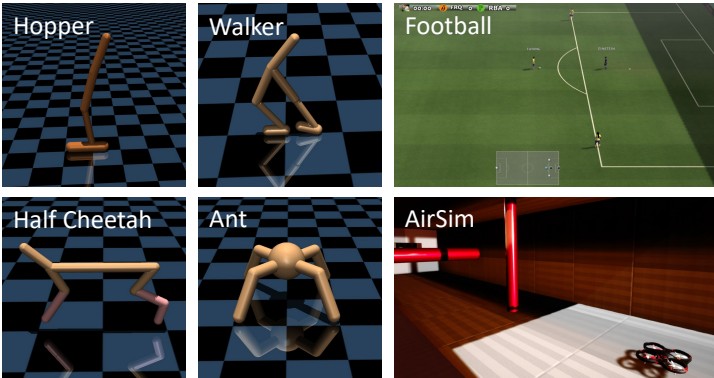

Figure 3: **MuJoCo** (left) is a set of popular continuous control environments with tasks of varying difficulty. **Google Research Football** (top right) is a novel RL environment where agents are trained to play football in an advance, physics based 3D simulation. **AirSim** (bottom right) is a new simulator built on Unreal Engine that offers physically and visually realistic simulations for both of these goals.

the agent can surpass the expert and achieve the near-optimal performance in sparse reward tasks, which corresponds to do not satisfy the Assumption 1. If the guidance is simply removed from the agent, in other words, the agent no longer distills knowledge from the guider, the bootstrap error and the uncertain reward signals can reduce the performance of agent policy. If we use the expert policy to restrict the agent, like other LfD methods, the agent policy will be excessively conservative, making it hard for the agent to achieve the near-optimal performance. Therefore, we address this issue by changing the learning objective of guider from expert to the agent, which will reduce the KL-divergence between $\pi_b$ and $\hat{\pi}$. The policy improvement lower bound when Assumption 1 is not satisfied is given as follows:

**Proposition 2.** *For policy $\hat{\pi}$ and any policy $\tilde{\pi}$,*

$$J_R(\hat{\pi}) - J_R(\tilde{\pi}) \geq -\frac{3R_{\max}}{2(1-\gamma)^2}\sqrt{2D_{\mathrm{KL}}^{\max}(\hat{\pi}, \tilde{\pi})}, \tag{11}$$

*where $R_{\max} = \max_{s,a}|R(s,a)|$.*

*Proof.* Refer to the Appendix B.

The guider update towards the agent will reduce the KL-divergence item to improve the policy improvement bound in Eq (11).

The mutual update of the guider policy and agent policy is the core feature behind the HYPO algorithm. In this way, the agent can leverage a few suboptimal demonstrations for efficient exploration while avoding the negative impact of the low-quality data.

## 5  Experiments

In this section, we investigate whether HYPO can achieve near-optimal performance in extremely sparse reward environments by overcoming the restriction of imperfect demonstrations, and how the number and quality of the trajectories can influence the performance of HYPO. To comprehensively assess our method, we first perform an exhaustive evaluation of HYPO in MuJoCo (Todorov et al., 2012) with sparse rewards and Google Research Football (GRF) (Kurach et al., 2020) with huge policy space and only a sparse score reward. We also evaluate HYPO on an Unmanned Aerial Vehicle (UAV) [2] task based on the Unreal Engine and AirSim (Shah et al., 2018) to show the effectiveness of HYPO in addressing more challenging control tasks with high-fidelity. All the mentioned environments are shown in Figure 3.

---

[2]https://github.com/sunghoonhong/AirsimDRL

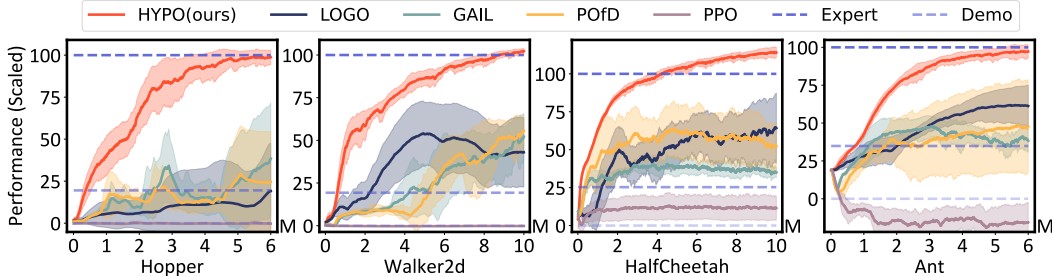

Figure 4: MuJoCo simulation results. The $x$-axis is the number of samples. The $y$-axis is the average episode return, which is scaled to make the expert achieve 100 and a random policy achieve 0.

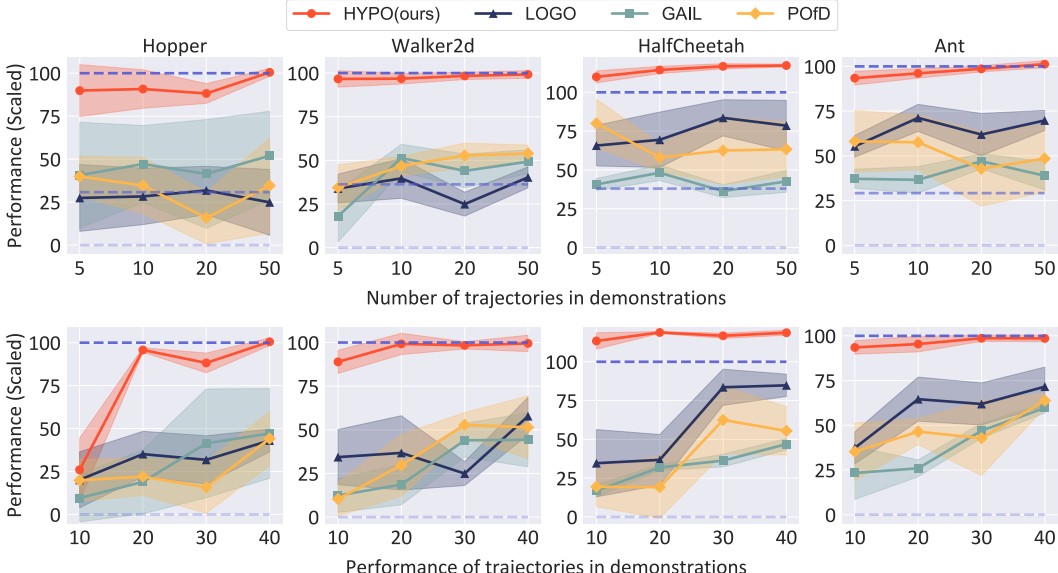

Figure 5: Final performance of the baseline algorithms that are trained with various number and performance of trajectories in demonstrations. HYPO is minimally affected by the quality of data.

## 5.1 MuJoCo Simulation

We first sparsify the built-in dense rewards on the MuJoCo platform for evaluating the methods in sparse-reward environments. Specifically, a reward of +1 is provided only after the agent moves forward over a specific distance. We compare HYPO to the following baselines: (1) **Expert**, which applies PPO using the original dense rewards to achieve the optimal return; (2) **Demo**, which is a suboptimal expert at the early stage of training; (3) **PPO**, which directly trains PPO with sparse rewards; (4) **GAIL** (Ho & Ermon, 2016), which uses a discriminator to provide a demonstration-guided reward for training; (5) **POfD** (Kang et al., 2018), which uses a weighted combination of the environment reward and the demonstration-guided reward; and (6) **LOGO** (Rengarajan et al., 2022), which merges a policy improvement step and a policy guidance step. All the demonstrations used for training are generated by **Demo**. Refer to Appendix C for more experimental details.

As shown in Figure 4, PPO fails to learn an useful policy in all environments due to the lack of guidance from demonstrations. GAIL can only achieve the level of the suboptimal expert policy since it can only mimic the policy that generates the demonstrations. POfD can attain a higher return than GAIL in HalfCheetah but a similar return in other tasks. LOGO can achieve relative better performance due to its decaying trust region that constrains the policy to gradually get rid of the influence of suboptimal data. However, the conservative policy can still impede the learning efficiency during the later learning process. As expected, HYPO outperforms the above methods in all environments by a large margin, which fully validates the effectiveness of HYPO in learning

near-optimal performance in extremely sparse reward environments by overcoming the restriction of imperfect demonstrations. We also investigate the influence of the number and quality (in terms of cumulative return) of demonstrations on the final performance. As shown in Figure 5, HYPO is capable of achieving far better performance than other methods, even with a small number of imperfect trajectories. Refer to Appendix F for more details about the performance of LOGO.

## 5.2 Google Research Football and AirSim Simulation

The GRF task is to control a single player to cooperate with teammates to break a specific defensive line formation and score. This environment provides two type of reward settings, i.e., the sparse score reward, and the dense checkpoints reward. AirSim is a high-fidelity environment where we need to control an UAV to fly to the terminal while avoiding all the obstacles. This task also provides the sparse reward setting for flying a specific distance and the dense reward setting for keeping a certain speed of forward flight. The expert baseline is trained in the dense reward setting, while the imperfect demo data is generated by a partially trained expert. The results in Figure 6 show that HYPO can achieve near-optimal performance in these challenging tasks with imperfect demonstrations.

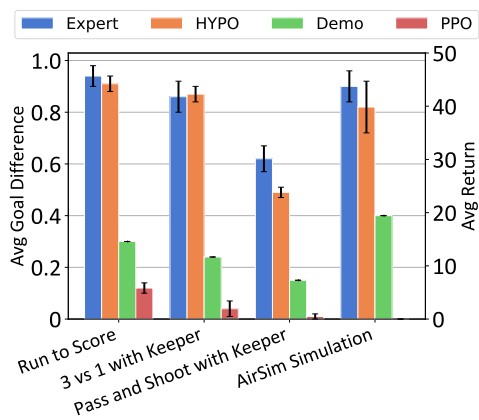

Figure 6: Results in GRF and AirSim. The left $y$-axis is for GRF and the right for AirSim.

## 6 Conclusion and Outlook

In this paper, we investigate how to accelerate agent online learning with imperfect demonstrations in sparse reward environments. We introduce HYPO, a novel RL algorithm that can attain near-optimal performance in sparse reward settings by avoiding the excessive conservative policy. The highlight of HYPO lies in its capability to learn an offline guider and an online agent, while updating these two policies mutually to help the agent learn from the demonstrations and environments more efficiently. Experiments in various environments including MuJoCo, Google Research Football and AirSim UAV simulation demonstrate that HYPO can greatly promote the learning efficiency in sparse reward tasks with imperfect data. Our future work is to extend HYPO to multi-agent scenarios when agents need to learn coordinated policies with sparse rewards. Moreover, we plan to investigate the potential of HYPO in more real-world applications, such as the autonomous driving, where sparse rewards are common challenges.

## Acknowledgments and Disclosure of Funding

This work was supported by an SYSU-ByteDance Research Project.

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
