# OpenReview forum: "Hybrid Policy Optimization from Imperfect Demonstrations"
_NeurIPS.cc/2023/Conference — NeurIPS 2023 poster_

### Official Review · Reviewer_5BQ6 · 2023-06-26

**Soundness:** 3 good
**Presentation:** 2 fair
**Contribution:** 3 good
**Rating:** 5
**Confidence:** 4

**Summary:**

This paper addresses the exploration challenge in RL by introducing imperfect demonstrations. These demonstrations are used to train an offline guider policy through imitation learning. Subsequently, the guider policy is employed to provide instructions during online training, thereby significantly enhancing the agent's exploration, particularly in the initial stages of training. Experimental results demonstrate that the proposed method outperforms certain baselines when faced with sparse rewards.

**Strengths:**

1. The motivation is clearly articulated.
2. The training approach for the guider policy is innovative.
3. The experimental results demonstrate that the proposed method outperforms several baselines significantly.

**Weaknesses:**

1. This paper introduces a novel approach for training the guider policy. However, the experimental results fail to demonstrate its superiority. It would be beneficial to conduct additional ablation studies to thoroughly analyze the proposed method and provide more comprehensive insights.
2. The comparison between the proposed method and baselines is limited to Mujoco experiments. It would be desirable to expand the comparisons to include other environments, allowing for a more comprehensive evaluation of the method's performance.

**Questions:**

Please refer to the weaknesses.

---

> ### Author Rebuttal · Authors · 2023-08-10
>
> We thank the reviewer for the insightful and valuable feedback. We explain the concerns point by point below.
>
> **Q1: About the superiority of HYPO and the ablation study.**
>
> A: We further elaborate on the superiority of our algorithm compared to other relevant methods and conduct more ablation experiments or sensitivity analysis to comprehensively evaluate the proposed method.
>
> About the superiority of HYPO: The most pronounced advantage of our algorithm is that, it can converge to near-optimal performance using only a **small amount** and **imperfect** (suboptimal & incomplete) demonstrations. While other related methods, such as LOGO, GAIL, POfD, require a large amount of demonstrations with high quality, and they are easy to suffer from the overly conservative policies, leading to a poor performance. These comparison results are provided on MuJoCo (e.g., Figure 5 in the main text). To provide a more comprehensive evaluation of our algorithm, we conduct more ablation study/sensitivity analysis. Please refer to the author rebuttal part and the figures in submitted PDF for more detailed results.
>
> **Q2: About more environments.**
>
> A: We have conducted more contact-rich manipulation tasks such as the [robosuite](https://robosuite.ai/) benchmark. Please refer to the author rebuttal and Table 1 (PDF in author rebuttal).

---

> > ### Comment · Reviewer_5BQ6 · 2023-08-18
> >
> > Thank the authors for the detailed answer. They have effectively tackled the majority of my concerns. As a result, I have made corresponding adjustments to my score.

---

### Official Review · Reviewer_2APk · 2023-06-26

**Soundness:** 3 good
**Presentation:** 3 good
**Contribution:** 2 fair
**Rating:** 7
**Confidence:** 4

**Summary:**

This work proposes a reinforcement learning algorithm to learn in a sparse reward environment with the help of data from a sub-optimal expert. The proposed algorithm, HYbrid Policy Optimization (HYPO) consists of three main parts, a discriminator, an offline guider, and an online agent. The primary idea is to use the sub-optimal expert data and discriminator loss, to train the offline guider.  The online agent interacts with the environment, and the offline guider is then used to guide the online agent using a BC loss.

**Strengths:**

The idea of using an offline guider is interesting, and the authors do a good job in explaining their algorithm and substantiate their claims with experiments.

**Weaknesses:**

While the paper has some interesting ideas, there are a few clarifications/weaknesses

1. The inclusion of the offline guider as part of the input to the discriminator lacks clarity. It would be beneficial for the authors to conduct an ablation study to better understand this aspect.

2. There appears to be a typo in equation 7,  it should likely be $\pi_b$ instead of $\pi$ (inferred from equation 6). Additionally, it is unclear whether the gradients flow through the discriminator (since $\pi_b$ is an input to the discriminator) when computing the gradient of equation 7.

3. How is the hyperparameter $\eta$ varied?

The authors claim that the offline guider is able to use information from the trajectories sampled from the online policy to learn from the agent (line 202 - 204). Which helps it attain a better perform than the original expert. The authors further substantiate this claim using proposition 2 (lines 267-270) and the design of the offline guider's loss function.
If this is the case,

4. Why is the value of C in equation 9 decayed? Shouldn't the training of the offline guider automatically bring $\pi_b$ closer to $\hat{\pi}$, thereby reducing the KL term to 0?

5. Isn't this decaying feature similar to LOGO (line 304-306), which the authors claim to result in conservative policies?

6. How is the value of C value decayed? What would happen if C is set to 1 and not decayed?

7. How is the KL divergence term calculated in equation 9? Is it an average KL?

8. The theoretical contributions of the paper seem minimal, primarily consisting of minor adaptations or direct reworkings of results from PofD and LOGO.

Suggestions that would further improve this work


9. The algorithm seems sensitive to the hyperparameters $\alpha$, $\eta$, and $C$. I would highly recommend the authors to conduct a sensitivity analysis, and provide the values used, and the decay methodology employed in the paper.

10. Inclusion of other baselines such as GAIL-PPO (use GAIL to train using sub-optimal data and then initialize the trained policy and perform PPO) and DAPG [1] will further substantiate the need for an offline guider.




Typos

Line 104: r is a function of state and action


[1] Rajeswaran, Aravind, et al. "Learning complex dexterous manipulation with deep reinforcement learning and demonstrations." arXiv preprint arXiv:1709.10087 (2017).


**Questions:**

See weaknesses.

**Limitations:**

I would encourage the authors to include a section on the limitations of their work.

---

> ### Author Rebuttal · Authors · 2023-08-10
>
> We thank the reviewer for the insightful and valuable feedback. We explain the concerns point by point below.
>
> **Q1: The ablation study of the input of discriminator.**
>
> A: We conducted additional ablation experiments to emphasize the significance of utilizing the offline guider as an additional signal to facilitate discriminator learning. Please refer to Figure 4 (PDF in author rebuttal) and the global author response for further details.
>
> **Q2: About $\pi$ in Equation 7 and the gradient flow.**
>
> A: In Equation (7), $\pi$ refers to $\pi_b$. We are sorry for this confusion and will modify it in the next version. Regarding computing $d(s,a,\log\pi_b)$, we indeed apply gradient clipping (in Lines 137-138 of `\HYPO\hypo\network\disc.py` in the submitted code).
>
> **Q3: About the hyperparameter $\eta$.**
>
> A: Please refer to the global author response and the Figure 1 (PDF in author rebuttal).
>
> **Q4: Why is the value of $C$ in Eq.9 decayed?**
>
> A: The primary purpose of decaying the coefficient $C$ is to gradually reduce the constraint imposed by the guider on the online agent. The online agent not only learns from the guidance of the guider but also interacts with the environment and learns from the rewards provided by the environment. When its performance is relatively poor or doesn't match the suboptimal expert's performance, it relies more on the guider's guidance. However, as the online agent learns and improves, surpassing the suboptimal expert, it increasingly requires exploration and learning from the environment. At this stage, if the guider continues to exert strong constraints, it could hinder the online agent's independent exploratory learning process, leading to excessively conservative strategies, similar to the results observed in the LOGO algorithm.
>
> We conducted additional experiments to validate this aspect, and the details can be found in global author rebuttal and the Figure 3 (PDF in author rebuttal).
>
> **Q5: About the conservative policy and the difference with LOGO.**
>
> A: First, the process of coefficient decay allows the guider's constraint on the online agent to gradually weaken. Consequently, the online agent's policy evolves from being passively guided to actively exploring. However, in the case of LOGO, the decay affects the policy's trust region $\delta$, leading to an increasingly conservative policy (smaller trust region means stronger constraint). This contrasts with HYPO's objective, which aims to achieve the opposite effect (i.e., achieving stronger constraint).
>
> Furthermore, HYPO is able to adopt this approach due to the dynamic improvement of the guider policy alongside the online agent. In LOGO, the pre-trained suboptimal expert policy remains static throughout, necessitating the constraint of the online agent's updates to prevent adverse effects stemming from the expert policy in the later stages of training.
>
> **Q6: About the hyperparameter $C$.**
>
> A: In all the experiments we have provided, the coefficient $C$ is decayed from 1.0 to 0.1. If $C$ remains constant without decay, the rate of improvement of the online agent would slow down, impeding the exploration efficiency of the online agent. This can be observed in the new added Figure 3 (PDF in author rebuttal).
>
> **Q7: How is the KL divergence term calculated in equation 9? Is it an average KL?**
>
> A: Yes, it indeed represents an average KL divergence. We initially record the action probability distribution of the guider policy and store it in the buffer of the online agent. When updating the actor of the online agent, we calculate the KL divergence between the computed online agent policy and the offline guider policy. The detailed calculation process can be found in the provided code, specifically in Lines 153-157 of `\HYPO\hypo\algo\hypo_ppo.py`.
>
> **Q8: About the theoretical contributions against to LOGO.**
>
> A: The primary distinction between the theoretical contributions of this paper and the LOGO approach lies in the fact that, in our work, after the online agent surpasses the suboptimal expert policy, the performance improvement lower bound is elevated through the learning of the guider policy, as opposed to the LOGO method which restricts updates via manually specified trust region parameters. Additionally, we provide theoretical analysis of the online agent policy's improvement, particularly in the later learning stages.
>
> **Q9: About the sensitivity analysis of the hyperparameters.**
>
> A: We have provided a comprehensive  sensitivity analysis of the hyperparameters. Please refer to the global author rebuttal and the PDF (PDF in author rebuttal).
>
> **Q10: About other baselines like GAIL-PPO and DAPG.**
>
> A: We conducted additional experiments and present the results in Figure 5 (PDF in author rebuttal).
>
> **Q11: About the limitations.**
>
> A: One of the limitations is the time complexity that we have mentioned in Appendix D. We will provide more discussion on this issue in the later version.

---

> > ### Comment · Reviewer_2APk · 2023-08-10
> >
> > I applaud the authors for their extensive rebuttal of the all reviewers comments. The new experiments add value to their work. I have a few additional comments,
> >
> > **1. Regarding the ablation study of the input to the discriminator**
> >
> > It would be helpful to see the impact of the ablation of $\log \pi_b$ on the final return, as in Fig 1(a) of the rebuttal. I would encourage the authors to include this on the final/next draft of their work.
> >
> >
> > **2. Regarding decaying $C$**
> >
> > I do understand the need for decaying $C$, and its clearly evident from Fig 3 in the rebuttal. Given these additional results, I am a little unsure of the final effect of *Adaptive target for Offline Imitation (section 4.2)*.
> >
> > I have modified my score accordingly, and wish the author my best!

---

> > > ### Author Response · Authors · 2023-08-11
> > >
> > > Thanks for the reviewer's additional comments. As you suggested, we will add the additional results into the next version of this work. Regarding the final effect of *Adaptive target for offline imitation*, if the online agent can learn a better policy than the suboptimal expert, the offline guider will learn a similar policy to the online agent since the weight $\mathcal{G}_{Agent}$ will get larger. On the contrary, if the online agent performance collapse (may occur if $C=0$, the policy may back to scratch in more complicated tasks), the offline guider will keep cloning the behavior that generates the demonstrations. In conclusion, the weights $\mathcal{F}$ and $\mathcal{G}$ decide the target for the offline guider according to the performance of the online agent.
> > >
> > > Sincere gratitude.

---

### Official Review · Reviewer_74Fn · 2023-07-03

**Soundness:** 2 fair
**Presentation:** 3 good
**Contribution:** 3 good
**Rating:** 5
**Confidence:** 4

**Summary:**

The paper introduces a method (HYPO) for RL using LfD, specifically in the setting where available demonstrations are suboptimal and/or incomplete. The method maintains a discriminator trained to differentiate agent-generated $(s,a)$ pairs from those available as demonstrations. The method further trains two policies simultaneously – an offline policy $\pi_b$ that is trained dynamically using behavior cloning on demonstration $(s,a)$ samples initially, and on those generated by the agent’s policy $\pi_{\theta}$ as the agent’s policy improves. $\pi_{\theta}$ is updated by minimizing a PPO-style, clipped-advantage loss function that also penalizes large deviations in $\pi_{\theta}$ from $\pi_b$ via a KL penalty. Experiments on sparse-reward versions of Mujoco environments show HYPO outperforms competing methods; sensitivity analyses demonstrate robustness to number/quality of demonstrations.

**Strengths:**

-	The idea of adaptive target BC to dynamically train the offline (guider) policy is novel and creative. Augmenting the PPO loss function in order to regularize the agent policy towards the guider policy is also an interesting contribution. Together, these ideas leverage available offline demonstration data in a useful way that enables faster policy learning.
-	All ideas are expressed clearly via adequate examples, illustrations, and experiments. The sensitivity analysis is rightly focused on the central premise of the paper—offline demonstration data and its suboptimality.


**Weaknesses:**

-	It has been emphasized that HYPO can learn from suboptimal and incomplete trajectories. While the sensitivity studies show adequate evidence for the former, it is not shown how HYPO can learn from incomplete demonstrations. A discussion of how HYPO’s formulation enables this and perhaps empirical verification could be helpful in this regard.
-	LOGO (Rengarajan et. al., 2022) considers a similar problem setting with suboptimal demonstrated behavior with sparse rewards. Their evaluations show that LOGO achieves expert-level performance on the Hopper and HalfCheetah environments. However, LOGO is shown to perform quite poorly in the results presented in the current paper leading to the feeling that the experiments here are perhaps skewed unfairly in favor of HYPO.
-	Confusing notation/lacking some consistency in annotating the guider/agent/demonstrator policies and the data samples that obtained from them. For example, it is easy to confuse the samples in $\mathcal{B}$ as coming from the guider policy $\pi_b$ (since both use ‘b’). Referring to them as demonstrator and demonstrations (rather than expert and expert demonstrations) might also help in easily differentiating between the two and make it easier to follow along.
-	(Minor) Having the algorithm in the main text of the paper (if it can be accommodated) would be helpful.


**Questions:**

-	How is $r_t(\theta)$ in Equation (9) computed? Specifically, is the probability ratio computed as $\frac{\pi_{\theta}^{k}(a|s)}{\pi_{\theta}^{k-1}(a|s)}$ as is done in the original PPO implementation (Schulman et. al., 2017)? Or is it computed as $\frac{\pi_{\theta}^{k}(a|s)}{\pi_{b}(a|s)}$ (where $k$ is the iteration.)
-	Do the current experiments employ a variant of LOGO that uses a different loss function or methodology from the original implementation (Rengarajan et. al., 2022)?
-	In Assumption 1, equation (4), if $\pi$ is random policy, is it always guaranteed that $ A_{\pi} (s,a_{e}) > A_{\pi}(s,a)$? (given that advantage $A_{\pi}(s,a)$ is computed under $\pi$ using $Q_{\pi}$ and $V_{\pi}$ which, for a random policy may be arbitrarily poor.)
-	(Minor) In practice, does training all three (the discriminator, guider policy and agent policy) simultaneously require significant hyperparameter tuning (given that the estimates of all three are likely to be extremely suboptimal and noisy initially) ?


**Limitations:**

Yes

---

> ### Author Rebuttal · Authors · 2023-08-10
>
> We thank the reviewer for the insightful and valuable feedback. We explain the concerns point by point below.
>
> **Q1: About the incomplete demonstrations.**
>
> A: In our method,  "incomplete" indicates that expert trajectories lack reward signals. Specifically, for offline learning, complete trajectories used for imitation typically consist of $(s, a, r, s^\prime)$ tuples. However, in HYPO, the trajectories used for learning lack the reward signal $r$ and consist only of $(s, a)$ pairs (the discriminator $d(s,a,\log\pi_b)$ in Eq.6 ).
>
> **Q2: About the performance of LOGO in this paper.**
>
> A: The experimental results of the LOGO algorithm presented in our paper were directly conducted using the publicly available LOGO [source code](https://github.com/DesikRengarajan/LOGO), without any modifications to its components. Our reproduced results align with the outcomes reported in their paper. The reason for the comparatively weaker performance of LOGO in the figures of our paper is attributed to the limitations imposed by the training samples. In the original LOGO paper, the x-axis reaches **1e7**, which consumes an enormous amount of samples. However, in our experiments, such as the Hopper environment, we only considered a maximum of 6M (x-axis is **1e6**) samples.
>
> Furthermore, although both LOGO and HYPO use suboptimal samples, LOGO's suboptimal samples reach over 60% of the performance of the optimal expert (e.g., in the Walker and HalfCheetah environments). In contrast, HYPO requires suboptimal samples that achieve just 20% or 10% of the optimal expert's performance, indicating that HYPO imposes less stringent requirements on expert trajectories. We will elaborate on this point further in the experimental section of the subsequent versions of the paper.
>
> **Q3: About the confusing notation.**
>
> A: We acknowledge the lack of uniform notation and consistency in certain aspects. We use the symbol $\mathcal{B}$ to indicate the 'buffer' of the online agent. We will change it to another symbol in order to clear the confusion.
>
> **Q4: About the algorithm pseudo-code.**
>
> A:  Thanks for the comment. We will consider including the algorithm into the main body of the paper in the next version.
>
> **Q5: The computation of $r_t(\theta)$ in Eq.9.**
>
> A: The computation of $r_t(\theta)$ in Equation 9 is conducted using the importance weight $\frac{\pi^k_\theta(a|s)}{\pi^{k-1}_\theta(a|s)}$, as is standard in the original PPO implementation. In HYPO, the online agent policy learns from the offline guider by exclusively distilling knowledge from the guider, without any additional specialized techniques. This simplicity allows HYPO to be conveniently implemented on PPO or other Policy Gradient (PG) methods.
>
> Regarding the latter form $\frac{\pi^k_\theta(a|s)}{\pi_b(a|s)}$, it could be employed if $\pi_b$ interacts with the environment and the trajectories are inserted into the replay buffer of the online agent. This approach bears resemblance to the [IRAT methods](https://proceedings.mlr.press/v162/wang22ao.html) as proposed by Wang et al. in 2022.
>
> **Q6: The implementation of LOGO.**
>
> A: Please refer to the response of Q2 above.
>
> **Q7: About the guarantee of $A_\pi(s,a_e) > A_\pi(s,a)$ in Assumption 1.**
>
> A: Intuitively, Assumption.1 implies that taking action acording to $\pi_e$ will provide a higher advantage than taking according to $\pi$ **since $\mathbb{E}_{a\sim\pi}[A_\pi(s,a)]=0$**. If the action is taken according to $\pi_e$ (e.g., $a_e$), the expectation $\mathbb{E}_{a_e\sim\pi_e}[A_\pi(s,a_e)]$ will be larger than 0 (the expectation is necessary, otherwise the assumption cannot be guaranteed).
>
> **Q8: About the hyperparameter tuning.**
>
> A: In fact, tuning the parameters of this method is particularly straightforward. In comparison to GAIL, our discriminator is no longer adversarial in nature, leading to significantly improved algorithm stability. Consequently, meticulous fine-tuning, as often required for GANs, is unnecessary. Although HYPO comprises three main components (the discriminator, the guider policy, the agent policy), the guider policy essentially functions akin to a Behavior Cloning (BC) task, involving minimal critical parameter adjustments. The parameters of the agent policy can be directly borrowed from the parameter settings in PPO, as we refrained from extensive parameter tuning specific to PPO.
>
> The primary parameters that require adjustment within the algorithm are the aforementioned $\eta$ (the positive class prior in Eq.6), $\alpha$ (the weight factor in Eq.8), and $C$ (the KL coefficient in Eq.9). Estimating these three parameters is remarkably straightforward, and we achieved impressive results with almost optimal settings upon initial configuration. Specific details regarding these parameters can be found in the beginning portion of the Author Rebuttal.

---

### Official Review · Reviewer_8Gvz · 2023-07-05

**Soundness:** 3 good
**Presentation:** 2 fair
**Contribution:** 3 good
**Rating:** 6
**Confidence:** 3

**Summary:**

This paper presents HYPO, a Learning from demonstration (LfD) method to effectively learn policy from a limited amount of imperfect demonstrations. At the core of the method are a hybrid discriminator to tell positive/negative state-action pairs through positive-unlabeled reward learning, an offline imitation learning policy to adaptively guide the policy learning to learn from the mix of expert and agent demonstrations, and an online policy learning extended from PPO to distill the knowledge from the offline policy. The method is evaluated on three benchmark datasets and shown to consistently outperform baselines.

**Strengths:**

1. The idea of improving learned policy beyond imperfect demonstration via treating policy rollout as unlabelled mix of positive and negative demonstration is insightful.

2. The method is tested on both simple and complex control tasks with low-dim and high-dim observation spaces. The learned policy can surpass the demonstration performance. The empirical performance of method over baselines is significant and convincing.

3. The ablation study on the number of trajectories is interesting and shows the robustness of the approach even on a few imperfect expert trajectories.

**Weaknesses:**

1. The exposition of the paper can be improved. The method part is not clear.

    a) In section 4.1, the authors use the word **agent policy** to refer to both online agent policy and offline agent policy.

    b) What is the discriminator's robust mean in section 4.2? and how $\pi_b$ improves the discriminator's robustness is unclear (the corresponding part in Appendix is hard to follow).

2. The necessity of the offline policy is not clear.

3. More ablation study of the hyperparameters is necessary.

**Questions:**

1. Are the agent demonstration $\mathcal{B}$ sampled from online agent or offline agent? From the context, it seems that they are sampled from online agent. However, in the discriminator loss function (Eq (6)), offline BC policy is used to provide the action probability $\pi_b(a|s)$. Could the author explain the mismatch here? Also will the inclusion of $\pi_b$ in Eq (6) result in a discriminator partially overfit to the $\log \pi_b$ and ignore $s, a$ (at least for the samples from $\mathcal{B}$)?

2. What is the exact schedule of $\eta$ and the value of $\alpha$? How is the algorithm sensitive to them? If so, is there empirically easy way to determine them?

3. What is the insight to have both an offline imitation learning policy and an online RL+imitation policy? Can we skip the offline imitation learning policy and only keep the online policy learning by modifying online policy update reward (Eq(9)) to include the dynamic imitation loss of Eq. (7)?

4. What is the secret sauce that the authors think is the key to the success of improving upon demonstrations? The learning of the discriminator and the offline policy is unware of which trajectory produced by the learned agent is positive or negative since it purely relies on a positive class prior $\eta$. Is it possible that the discriminator learns a wrong quality assignment thus confuses the learning of the offline policy with wrong $\mathcal{G}$?

5. The author claims the learned policy is near-optimal for several times across the main text, which is a too strong claim without strict evidence and might mislead the reader and oversell the paper.

6. How is the method's performance on more contact-rich manipulation tasks? Such as the [robosuite](https://robosuite.ai/) benchmark?

---

> ### Author Rebuttal · Authors · 2023-08-10
>
> We thank the reviewer for the insightful and valuable feedback. We explain the concerns point by point below.
>
> **Q1: About the presentation and confusion of notations.**
>
> A: In the paper, the **guider** is trained using **offline imitation learning**, and thus, we use the term offline imitation policy (guider) to indicate it. The **agent** is trained using **online reinforcement learning**, and we use the online learning policy (agent) to indicate it. We will proofread the paper thoroughly in order to keep consistent of these terms.
>
> **Q2: About the  robustness of the discriminator.**
>
> A: The robustness of the discriminator refers to its ability to accurately differentiate between positive and negative samples even when there is a significant variation in the performance of the input samples. This is particularly relevant since the performance of the online policy is consistently improving, causing substantial disparities in the quality of the input samples. We make $\pi_b$ challenge the discriminator $d$ by doing the opposite to minimizing $\mathcal{L}_d$, and this can be seen as minimizing the worst-case error ([Carlini et al., 2019](https://arxiv.org/abs/1902.06705); [Fawzi et al., 2016](https://proceedings.neurips.cc/paper/2016/hash/7ce3284b743aefde80ffd9aec500e085-Abstract.html); [Goodfellow et al., 2015](https://arxiv.org/abs/1412.6572)).
>
> **Q3: About the necessity of the offline policy.**
> A: The offline policy is a crucial component of our method and plays a critical role in achieving the final performance:
>
> 1. In the initial training stage, the offline policy emulates the behavior of the expert policy and guides the learning of the online policy. Without the guidance of the offline policy, the online policy would struggle to learn meaningful policies in environments with such sparse rewards.
> 2. Throughout the training process, the offline policy dynamically selects imitation targets. As the performance of the online policy surpasses that of the expert, the offline policy adjusts to imitate the online policy to stabilize the online policy's updates, preventing the emergence of overly conservative policies.
>
> **Q4: More ablation study of the hyperparameters.**
>
> A: We conducted additional experiments to perform ablation studies on the hyperparameters. The results of these experiments can be observed in Figures 1-3 (PDF in author rebuttal).
>
> **Q5: About the $\pi_b(a|s)$ in Equation 6.**
>
> A: First, $\mathcal{B}$ is sampled from the online agent policy. Regardless of whether $(s,a)$ comes from $\mathcal{B}$ or $\mathcal{D}$, the discriminator consistently treats $\pi_b(a|s)$ as an additional signal to aid in its training. Specifically, if $(s,a)$ coms from $\mathcal{B}$ (online agent), the expected value of $\pi_b(a|s)$ is higher due to the policy generating $(s,a)$ and $\pi_b$ being closely aligned with each other, resulting in a small KL divergence between them. Conversely, if $(s,a)$ comes from $\mathcal{D}$ (expert demonstrations), the expected value of $\pi_b(a|s)$ is lower since $\pi_b$ significantly deviates from the expert policy responsible for generating the demonstrations. Consequently, the input to the discriminator in Equation 6 includes $(s,a,\pi_b)$.
>
> **Q6: About overfitting to $\log\pi_b$ in discriminator.**
>
> A: If the discriminator in HYPO overfits to $\pi_b(a|s)$, it can indeed impact the accuracy of the discriminator. In order to demonstrate the role of $\pi_b$, we have provided a comparison of the discriminator's performance during training. See the results in Figure 4 (PDF in author rebuttal).
>
> **Q7: About the hyperparameters $\eta$ and $\alpha$.**
>
> A:  Please refer to the author rebuttal and Figures 1,2 (PDF in author rebuttal) for details.
>
> **Q8: About the insight of using both an offline imitation learning policy and an online RL policy.**
>
> A: The key insight of our method is to harness the guidance from the offline imitation policy to aid the learning of the online policy while avoiding an excessive conservative impact and facilitating convergence. If we solely rely on online agent imitation learning without the guider  (offline imitation policy), the online policy can be excessively constrained by the expert demonstrations, leading to overly conservative updates and failure to converge to an approximate optimal level.
>
> Regarding the combination of updates in Equation 9 with Equation 7, we acknowledge it as a promising avenue that could be explored in future research. However, it appears unfeasible within the existing HYPO framework.
>
> **Q9: About the secret sauce of improving upon demonstrations.**
>
> A: The discriminator can learn to distinguish trajectories solely relying on the original loss function (3) in this paper. The PU-Learning and $\eta$ are introduced to mitigate the overfitting problem, by re-weighting the losses for positive and unlabeled data. Therefore, the discriminator can learn to distinguish these trajectories correctly. The accuracy of the discriminator is shown in Figure 4 (PDF in author rebuttal).
>
> **Q10: About the claims of near-optimal performance.**
>
> A: Indeed, the usage of 'near-optimal' might appear too strong. The reason behind this claim in the paper was the observation that the convergence performance closely aligns with that achieved under dense rewards. We will revise this claim in the next version of the paper.
>
> **Q11: About the performance of HYPO on robosuite.**
>
> A: To evaluate the algorithm's performance in robosuite environments, we conducted additional experiments, and the results of which are illustrated in Table 1 (PDF in author rebuttal) and the last section in author rebuttal.

---

> > ### Comment · Reviewer_8Gvz · 2023-08-10
> >
> > Thank the authors for the detailed answer. The authors have addressed most of my concerns. Though it is still unclear whether it is necessary to have the offline policy since we can probably combine Eq 9 with Eq 7 and train online policy only, the experiment performance of the proposed algorithm and its new evaluation on robosuite manipulation task indicates that it can be a valuable work to the community. I have updated my score. In the revision, I hope the authors can proofread the manuscript and significantly improve the presentation of the paper.

---

### Official Review · Reviewer_8vGE · 2023-07-09

**Soundness:** 3 good
**Presentation:** 3 good
**Contribution:** 3 good
**Rating:** 7
**Confidence:** 4

**Summary:**

The goal of this work is to develop an approach for RL in the sparse reward setting, wherein the RL agent is directed by a guidance policy that provides direction to the online learner.  While this approach has been used before, the main contribution of this work is on developing a guidance policy that utilizes both offline data gathered by a behavior policy and online data gathered by the learning agent.  KL-Regularization with respect to this guidance policy is used to guide the learning agent, with the level of regularization being reduced as the learning proceeds so that the policy learned can be optimal.   Some theoretical justification is provided, following in much the same manner as LOGO (Rengarajan et al. 2022) that shows that the guidance procedure will extract an advantage from the offline data if it is possible to do so.

**Strengths:**

+ Interesting idea on how to bring in guidance using both offline and online data.

+ The guidance policy should smoothly go from a pure BC of the data to being identical to the online policy.  This should help in reducing excessive dependence on the guidance policy as in LOGO.

**Weaknesses:**

- The KL regularization coefficient is indicated as decaying with time.  If the guidance policy truly converges to the online policy, the coefficient should not actually need decaying at all.   Indeed this is much the same as the excessive conservative behavior that the authors indicate is a problem with LOGO.

- It is also unclear how exactly the regularization is undertaken or how the coefficient is decayed.  Is this tuned up by hand for each environment?  That would be an unfair comparison with the other approaches where the decay rate of guidance is implicitly chosen by the algorithm itself.

- In some of the experiments (HalfCheetah), HYPO does better than the upper bound (PPO trained with a dense reward).  How is this possible?

Smaller issues:
- The analytical part is essentially identical to LOGO, although that is not a major contribution of the paper.

- The second set of figures on page 5 have the x-axis missing the M (which I guess stands for million samples).  They can't possibly train in 40-50 samples

**Questions:**

Please see negatives above.  I think this is an interesting idea, and would like to see your responses to the decay question in particular.

**Limitations:**

Yes.

---

> ### Author Rebuttal · Authors · 2023-08-10
>
> We thank the reviewer for the insightful and valuable feedback. We explain the concerns point by point below.
>
> **Q1: About the decaying of the KL regularization coefficient.**
>
> A: The primary purpose of the regularization term here is to facilitate knowledge distillation from the guidance policy to the online policy. We opt for a decaying approach of the regularization term to avoid introducing an overly conservative policy to the online agent. As the performance of the online policy surpasses that of the expert, the guidance policy converges to the online policy gradually, and thus there is no need to decay the coefficient anymore. Two aspects, however, require further clarification:
>
> 1. First, HYPO does not induce an excessively conservative policy akin to that observed in LOGO. In LOGO, the guidance is provided through pre-trained policies using Generative Adversarial Imitation Learning (GAIL), indicating a fixed proficiency level of the guidance policy. This, in turn, imposes significant limitations on the subsequent enhancement of the online policy.
> 2. Although the coefficient of the regularization term remains constant during the later stages, it remains a relatively low value, specifically 0.1 for all environments. This is informed by the necessity to avoid potential performance collapse due to unstable exploration under sparse rewards, as illustrated in Figure 3 (the PDF in author rebuttal).
>
> **Q2: About how exactly the regularization is undertaken or how the coefficient is decayed.**
>
> A: The regularization coefficient linearly decays from 1.0 to 0.1 across all environments in the paper,  with a step size of 2M. We do not manually adjust this coefficient for each environment, since we only need to ensure that the regularization term loss and actor loss are of the same magnitude, similar to the control of entropy loss coefficient.
>
> **Q3: About the results that HYPO does better than PPO trained with dense reward.**
>
> A: In certain experimental environments, we indeed observe that agents trained through imitation learning can outperform the expert policies trained under dense rewards. This result can be also observed in the following cases: 1) Figure 1 in the LOGO paper; 2) Figure 3 in the POfD paper; and 3) Figure 1 in the GAIL paper. We hypothesize that this result can be attributed to the following reasons:
>
> 1. Mismatch in Reward Signals: During the training of HYPO, only the primary long term reward signal, i.e., forward movement, is employed. However, in the original HalfCheetah environment, additional dense rewards such as control cost are present. While in specific environments, these extra reward signals might aid agents in mitigating the challenges posed by sparse rewards, expediting convergence, there are still scenarios when these additional signals might not be consistent with the long term performance.
>
> 2. Approximate Optimal Policies: Policies that converge under dense rewards may not necessarily represent optimal solutions. The convergence of Policy Gradient (PG) methods like PPO is influenced by several factors, such as exploratory aspects (entropy). Hence, different combinations of parameters and components might impact the convergence performance of the algorithm in various ways. While we have selected a set of reasonably suitable parameters and maintained consistency between the training of PPO and HYPO, HYPO incorporates some additional parameters compared to PPO. It is possible that, under the combination of these parameters, HYPO manages to explore more efficiently than PPO.
>
>
> **Q4:  About the analytical part compared with LOGO.**
>
> A: The difference of the theoretical analysis between HYPO and LOGO lies in the fact that the guidance policy imitates both the expert and the agent to provide guidance. Specifically, in the initial stages, the analysis is similar to LOGO since the online agent is learning from scratch and needs to get guidance from the demonstrations. Once the online policy surpasses the performance of the demonstration, the offline guider policy aligns itself with the online policy to enhance the policy improvement bound by decreasing $D_{KL}$ :
>
> $$
> J_R(\hat\pi)-J_R(\tilde\pi)\ge -\frac{3R_{\rm{max}}}{2(1-\gamma)^2} \sqrt{2D^{\max}_{\rm{KL}}(\hat\pi,\tilde\pi)}. \tag1
> $$
>
> In LOGO, the authors manually control the trust region $\delta_k$ for policy updates:
>
> $$
> J_R(\pi_{k+1})-J_R(\pi_k)\ge-(\sqrt{2\delta}{\gamma\epsilon_{R,k}}+3R_{\max}\delta_k)/(1-\gamma)^2, \tag2
> $$
>
>  which results in overly conservative policies and unstable convergence.
>
> **Q5: About the x-axis in Figure 5.**
>
> A: The x-axis in Figure 5 does not lack 'M'. The x-axis indicates "the number of trajectories in demonstrations", with each trajectory containing approximately 1k transitions (samples). These samples are indeed sufficient for HYPO's training. As the offline guider in HYPO does not interact with the environment and is responsible solely for providing guidance to the online policy, issues such as distribution shift (out of distributions) commonly seen in offline learning due to inadequate sample size can be readily addressed. This serves as one of the major advantages of HYPO – it requires only a **small amount** of **suboptimal** trajectories, facilitating easy deployment and efficient application.

---

> > ### Comment · Reviewer_8vGE · 2023-08-20
> >
> > Thank you for your responses.  I have increased the score appropriately.

---

### Author Rebuttal · Authors · 2023-08-10

We would like to express our sincere gratitude for the thorough review and valuable feedback on our paper. The reviewers' insights and suggestions are quite helpful to improve the quality and clarity of our work. In this author rebuttal, we conduct additional experiments about the **ablation study**, **sensitivity analysis**, more **baselines** and more **environments**, which are the major common concerns of the reviewers. We provide brief analysis of these results below.

**1. Sensitivity analysis of the hyperparameters**

There are three main hyperparameters in our method HYPO: the positive class prior $\eta$, the weight factor $\alpha$, and the regularization coefficient $C$. During the development of our algorithm, we did not tune these hyperparameters much, as they are easy to estimate. For example, $\eta$ is set to 0.5 in most of the PU-Learning tasks, while we set it to [0.2, 0.8] due to the performance improving of the online agent policy. As for $\alpha$, we need to ensure $\mathcal{F}_{\text{Expert}}>0$ whith the domains of $\eta$ and $d$. Therefore, the value of $\alpha$ must be greater than 5.0. Finally $C$, which can be set like the entropy coefficient, needs to ensure that the losses are within the similar order of magnitude.

First, Figure 1 (PDF in author rebuttal) illustrates the impact of parameter $\eta$ on our method. In this figure $\eta$ indicates the initial value, and linearly increases to the end value of $1-\eta$. $\eta$ primarily affects the updates of the guider, since it determines the weight of the positive samples that come from the online agent. But the online agent performance is rarely affected by the value of $\eta$ (Figure 1a). Second, Figure 2 (PDF in author rebuttal) illustrates the impact of parameter $\alpha$ on HYPO. Similar to $\eta$, $\alpha$ also primarily affects the updates of the guider.  However, the offline imitation policy can dynamically adjust these two weights of $\mathcal{F}$ and $\mathcal{G}$, which means that bigger $\mathcal{F}$ can lead to bigger $\mathcal{G}$ (Figure 2b, 2c). In this way, the weight factor $\alpha$ rarely affect the learning of the guider. Therefore, the agent's performance is not sensitive to $\alpha$ (Figure 2a). Finally, Figure 3 (PDF in author rebuttal) illustrates the impact of parameter $C$ on HYPO. In this figure the $C_{min}$ indicates the end value of $C$ (all decay from 1.0). If $C$ has a large value, such as 0.5 or 1.0 (not decaying), the performance of the online agent can be affected due to the overly restrictions (Figure 3a). However, $C$ also can not be too small, such as 0. In this case, the KL between the offline guider and the online agent would be very large, leading to collapse in the agent's performance.

**2. The ablation study of the input in discriminator**

We compare the accuracy of distinguishing the expert data and the agent data to study the necessity of $\log \pi_b$ in the discriminator. As illustrated in Figure 4 (PDF in author rebuttal), introducing $\log \pi_b$ in the input of discriminator is beneficial for accurate discrimination.

**3. New baselines: GAIL-PPO and DAPG**

We add new baselines and test them in the MuJoCo environments (e.g., HalfCheetah, as shown in Figure 5 (PDF in author rebuttal)). These new baselines include: 1) **GAIL-PPO**, which trains GAIL to get a suboptimal performance and then initialize the trained policy and perform PPO; and 2) **DAPG**, which warm starts Natural Policy Gradient (NPG) algorithm using BC, and fine tunes it online using behavior data in a heuristic manner.

**4. New environment: robosuite**

In order to more comprehensively evaluate the performance of HYPO, we conduct additional experiments in [robosuite](https://robosuite.ai/) environment. We choose two easy tasks (Block Lifting, Door Opening) and a hard task (Two Arm Lifting) to test our method. First, we use PPO benchmark to train an expert in the dense reward setting (use the 'reward shaping' option). Then we use a partially trained expert that is still at a sub-optimal stage of learning to provide the demonstrations. Finally we implement HYPO, GAIL and POfD in these tasks without using 'reward shaping' to compare the performance, as shown in Table 1 (PDF in author rebuttal).

---

### Decision · Program_Chairs · 2023-09-21

**Decision:**

Accept (poster)

**Comment:**

This paper proposes an approach for RL in the sparse reward setting, wherein the RL agent is directed by a guidance policy that provides direction to the online learner. While this idea has been studied before, the main contribution of this work is on developing a guidance policy that utilizes both offline data gathered by a behavior policy and online data gathered by the learning agent.


The review scores are 7, 7, 6, 5, 5 and the average score is 6.0. The reviewers are generally positive about the novelty of the algorithmic contribution, clarity of presentation, and technical soundness. The main concern expressed by multiple reviewers was about the similarity and the lack of detailed comparisons with the LOGO algorithm (Rengarajan et al. 2022). The authors have given detailed response these comments, and the reviewers have accepted their explanations.

For the final version, please make sure that you include the comments and suggestion by the reviewers.

Thanks,

AC